# The Effectiveness of Computer Aided Video Modeling in Teaching Basic Basketball Movements to Individuals with Down Syndrome

**DOI:** 10.3390/children10010153

**Published:** 2023-01-12

**Authors:** Aslan Aydogan, Mukaddes Sakalli Demirok

**Affiliations:** Special Education Department, Atatürk Education Faculty, Near East University, Nicosia 99138, Cyprus

**Keywords:** basketball, down syndrome, video modeling, skill, technology

## Abstract

In this study, the effectiveness of the video modeling method in teaching basic basketball skills to students with Down syndrome was examined. Four students with Down syndrome, who were studying at the disability free living center in Dolayaba, participated in this study. The ages of the students diagnosed with Down syndrome were as follows: two of them were 13, the others were 14 and 16. In the study, the effectiveness of video modeling in teaching basic basketball skills was evaluated using the multiple probe model, one of the single-subject research methods. The experimental process of the study consisted of collecting baseline data, daily, conducting instructional sessions, maintenance and generalization stages. The findings of the study showed that video modeling was effective in teaching basic basketball skills to students with Down syndrome and that they maintained the skills in interpersonal and environmental differences after the end of the instruction. When the opinions of the basketball teacher and the students were evaluated, it was determined that the students’ self-confidence and peer relations were strengthened thanks to their active participation in the lessons. The students were not bored because they stated that they found the lessons interesting and fun.

## 1. Introduction

Down Syndrome (DS): It is a type of physical, mental, and functional disability caused by an extra copy of the twenty-first chromosome. [1,2]. Its clinical symptoms were first identified by Langdon Down in 1866 [3]. It is the most common type of chromosome anomaly in humans. People with Down syndrome have some degree of intellectual disability—usually in the mild to moderate range. The chromosome disorder affects many organs (heart problems, hearing deficiencies, etc.).

Just like individuals with a typical development, special-needs individuals need to learn a variety of skills required by societal life. In acquiring skills such as self-care, everyday routine skills, shopping, traveling, etc., the objectives of the institutions continue to depend on the regulation of content and the delivery of appropriate methods or approaches to content [4,5]. One of those approaches is a sport. This is because participation in physical activity works in the name of revealing one’s different identities and roles: it changes their perception and allows them to gain different perspectives; it stimulates the need to be in a group and the sense of being part of a team; it recognizes incompetence and takes precautions and improves [6]; and it contributes to an individual’s relief by fighting stress [7]. Here is how Malina and Cumming have explained some of the possible benefits of sports engagement; the effects of maturation are regular physical activity, self-concept or self-worth effects, social competence, and moral development which allow for improved quality of life [8].

Research has shown that fair play, sportsmanship, and knowledge of moral development of sporting individuals have an impact on the moral character development from sport, when continually taught [9]. Mobility, exercise, and sport are a means of community participation for individuals with disabilities and the development of some basic motor skills, and physical characteristics. Promoting this tool before and after special education is quite important [10]. Physical activity studies, while important for all individuals, are of greater importance to individuals with disabilities [11,12,13,14].

Studies have reported that children and youth with DS lead a more sedentary lifestyle than their peers with normal development and as a result, they are more affected by health problems such as overweight and obesity [15,16], and children with DS are more likely to be overweight and obese than their peers with normal development [16,17]. Physical activity studies are also critical, especially for individuals with DS, where the risk of obesity is higher than in other mentally disabled peers. Major factors that adversely affect the involvement of individuals with DS in physical activity are muscle hypotonia, connective tissue laxity, the tendency to mild or moderate obesity, an inadequately developed respiratory–circulatory system, and short height [18]. In addition, low motor skills such as balance and grip also negatively affect the participation of individuals with DS in physical activity and sports [19,20,21].

Individuals with disabilities can learn individual or team sports with appropriate training and support, enjoy them and have a healthy body [22]. One of these team sports is basketball. Basketball is an effective team sport both mentally and physically; starting with childhood and youth, with targeted research, it improves physical strength characteristics such as endurance, strength, speed, skill, and mobility in a usual manner and brings them to a higher level. The need to implement technical and tactical elements in the position of sudden changes in the game is also an important factor in the development of functions such as coordination and reaction. At the same time, the overall development of the organism will help eradicate the disease and produce solid organisms. It is easy to identify the personality traits of athletes in the game and take the prescribed training measures on time, enabling athletes to achieve an informed level of discipline, collective thinking, and to practice and participate in their work in a team discipline. Basketball has a very positive impact on individual personality training because the quality of the game is suitable for people of all genders and ages. It includes traits such as struggle, courage, integrity, cooperation, fairness, and self-reliance. It positively impacts individual psychology and social behavior, and brings a beneficial personality to society [23]. In basketball, basic movements are shooting, pass-making, and dribbling.

Shooting: Attacking player aims to throw the ball directly into the basket. Shooting is used to conclude an attack [24].Pass-making: It is the method that the player uses to transfer the ball to his teammates in different ways with one or both hands [24].Dribbling: The ball bounces off the ground when it hits the ground. This is performed with the left or right hand. The ball is bounced and moved from place to place, by running or walking. It is called basketball dribbling [24].

In sporting skills, as in all fields, when teaching special-needs individuals a new skill, process, or concept, students can have a successful learning life depending on the teaching method or approach the teacher uses. Teaching methods are used to teach skills and concepts to individuals with special needs, and the skills or concepts to be taught have advantages and limitations over their characteristics [25]. New methods are being developed in studies for effective teaching, and adaptations are being made to make the methods more effective and efficient [26]. Looking at the field writing on effective teaching, there are many different practices or approaches [27]. One of them is the teaching method of becoming a model with a video. Education teaching by video model is one of the underlying scientific applications whose effects are proven through experimental research. This practice arose from the hypothesis that students learn from social learning theory by observing others [28,29].

There are two fundamental processes of learning by observation. These are the abilities to model and imitate. This process is highlighted in the field as a whole. The student exhibits the target behavior, and this becomes a clue for other students [30].

The literature shows that researchers worldwide have been extensively investigating the effectiveness of video modeling in recent years [31]. Van Laarhoven, Laarhoven-Myers and Zurita have conducted a study looking into the effectiveness of teaching with the video model, offered with the help of a pocket computer, in teaching job skills to individuals with mental deficiencies [32]. As a result of the study, the video model provided with the help of a pocket computer found that this teaching was effective in teaching business skills. Bidwell and Rehfeldt taught adults with mental disabilities the skills to make coffee, serve coffee, and sit with peers [33]. Allen et al., have conducted a study on the effectiveness of teaching by video model in acquiring vocational skills for young people with autism [34].

Research has shown that video modeling is effective in the education of individuals with autism and intellectual disabilities. However, the fact that there are few studies on both gaining sports skills and modeling with video, in studies on individuals with Down syndrome, reveals the importance of this research.

### Purpose of Study

This study aims to determine the effectiveness of teaching to model with video in teaching basketball basic movement skills to individuals with Down syndrome. For this purpose, the research sought answers to the following questions.

Could teaching by video model give students with Down syndrome the ability to shoot from viewing basic basketball skills?Could teaching by video model give students with Down syndrome the ability to pass upon viewing basic basketball skills?Could teaching with the video model give students with Down syndrome the ability to dribble from viewing basic basketball skills?With the video model, can teaching be provided to students with Down syndrome, if they master the basic movements of the basketball skills, to help them generalize the skills (with different people and in different environments)?If video modeling teaching provides basic basketball skills, can the permanence of this outcome be ensured ten days and twenty days after the learning ends?Is teaching with video modeling practical on the social development of students with Down syndrome?

## 2. Materials and Methods

In this chapter, we cover the research model, the dependent and independent variables of the research, the data collection tools used, teaching materials, application reliability, experimental process, information on the collection and scoring of data, analysis, and interpretation of data, and calculation of trustworthiness among observers.

### 2.1. Research Model

In this study, in order to determine the effectiveness of video model instruction in teaching basic basketball skills to students with Down syndrome in the acquisition, maintenance, and generalization of these skills, a multiple probe between subjects with probe phase, one of the single-subject research methods, was used.

Single-subject research is a research method that involves time series analysis based on repeated measurements or observations before, during, and after an experimental intervention on a subject, in order to reach a basic causal relationship [35,36,37]. The series of measurements or observations before the intervention serves as an indicator of a person’s behavior or acquired skill after the intervention [38].

One of the most important functions of single-subject research is that it helps to demonstrate that the changes in the target behavior are not due to other factors, but only due to the practice [39].

### 2.2. Implementation Stages of Video Teaching

Before the implementation of video instruction, the researcher told the students that they would watch a video on the computer and then the videos were shown to the students. In the videos, the modeling stages of each skill were shown. After the modeling stage, the video was stopped, and instructions were given to the student. In the third stage, the independent practice stage, the student was asked to perform the skill he/she saw and experienced in the video. The same steps were followed for each skill.

Teaching sessions lasted approximately 20–25 min. In the independent practice phase, social reinforcement was provided by using expressions such as “Well done, now it is done” when the student responded correctly.

### 2.3. Dependent and Independent Variable

The dependent variable of this research is basic basketball skills. The independent variable of research is the teaching to model with video.

### 2.4. The Setting and Materials

The study was conducted in the basketball hall of the Center for Accessible Life in Dolayaba, where the students received education. The baseline, instruction, maintenance, and generalization sessions of the study were conducted in the same environment. There were four fixed hoops, two free-standing hoops, twenty-five funnels, and thirty basketballs in the basketball hall. Video images prepared for the target skill, 8 Inc. tablet computer, 13 Inc. laptop computer, computer cable, camera, tripod, and data collection forms were used in the study.

### 2.5. Experimental Process

The experimental process included baseline session, instructional, mass probe, daily probe, maintenance and generalization sessions for each subject.

### 2.6. Participants

Four subjects participated in the study, Yusuf and Mikail 13, Ismail Can 14, Salih Can 16 years old. All subjects have been diagnosed with Down syndrome since birth and all subjects receive education in a special education and rehabilitation center.

### 2.7. Mass Probe Sessions

Probe sessions were held in the form of mass probe sessions and daily probe sessions. Mass probe sessions were conducted at the commencement level and simultaneously in all subjects after the end of teaching in each subject. The daily probe sessions were implemented at the beginning of instructional sessions in all topics, excluding the primary education session, and only for the subject in teaching.

Five mass probe sessions were held to determine the pre-teaching performance of subjects and to determine the performance of subjects after they perform at a benchmark level and after stable data is obtained. The first of the mass probe sessions was held to collect kick-off level data.

Mass probe sessions were held one-on-one with students at the location of the practice. In the first subject, three consecutive sessions were continued until stable data were obtained. After stable data were obtained, a training session was initiated for the first student. When the criterion for the first subject was fully met, the second probe session was conducted for all subjects. Three consecutive sessions with the second subject were continued until stable data were obtained, and after stable data were obtained, the instruction session with the second subject was started. The third probe session was conducted for all subjects when the second subject performed at the level of meeting the criterion. Three sessions with the third subject were continued until stable data were obtained, and after stable data were obtained, the training session with the third subject was started. When the third subject also performed at a level to meet the criterion, a fourth probe session was conducted for all subjects. The training session was continued for three sessions with the fourth subject until stable data were obtained, and after stable data were obtained, the training session was started with the fourth subject. When the last student performed at a level to meet the criterion, a final probe session was organized for all students.

### 2.8. Daily Probe Sessions

It was held before the instructional session on other days, except for the day when instructional sessions started in the daily probe sessions. Only the student’s performance data was collected.

### 2.9. Instructional Sessions

Training sessions were conducted three days a week and one session per day. During the session, students were taken to the hall where the intervention would take place. In the hall, the laptop was kept stationary on a table and the tablet was used to allow the practitioner to watch videos of the student standing in critical situations. The student was told “we will watch a video with you and you will do the same thing you see” and “if you do the same thing you see in the video, I will give you a reward that you will like”. Then the video was opened and watched. After the student watched the video, a verbal reinforcement such as “well done, you watched it very well” was given. After watching the video, the student was given an instruction such as “take the ball and do the same”. The successful behavior of the subject was verbally reinforced. When the student made a mistake, the student was told “let’s watch the video again”. After watching the video again, the student was again given the instruction “let’s do the same”.

### 2.10. Maintenance and Generalization Sessions

Maintenance sessions were held 10 and 20 days after the instructional was ended. Generalization sessions were conducted in the form of interpersonal training in different settings. Generalization sessions have been convened by the pupils’ sports teacher who works at the institution. It has thus been investigated whether they generalize the skill gained against different individuals.

### 2.11. Reliability

The research collected data on reliability and application across observers. For analysis of trustworthiness data between observers, the consensus/consensus + difference of opinion X 100 “formula was used. For application reliability, the formula “Observed practitioner behavior/planned practitioner behavior X 100” was used [40].

## 3. Results

This section includes findings on the effectiveness, maintenance, and generalization of basic basketball skills. The effectiveness, maintenance, and generalization findings chart shows the number of horizontal axis sessions, vertical axis daily probe sessions, and the right response percentages in generalization, instructional sessions. The data on the shooting skills of the subjects are given in Figure 1 below.

Yusuf showed an average performance of 15% in the pre-instructional process. At the end of the instruction and probe sessions, it was observed that he responded correctly to the target stimulus with an average of 90%. In the follow-up sessions, which were conducted ten and twenty days later, it was observed that he maintained 90%.

Ismail Can showed an average performance of 25% in the pre-instructional process. At the end of the instruction and probe sessions, it was observed that he responded correctly to the target stimulus with an average of 85%. In the follow-up sessions, which were conducted ten and twenty days later, it was seen that he maintained a level of 85%.

Mikail showed an average performance of 25% in the pre-instructional process. At the end of the instruction and probe sessions, it was seen that he responded correctly to the target stimulus with an average of 90%. It was seen that he maintained a level of 85% in the follow-up sessions, which were conducted ten and twenty days later.

Salih Can showed an average performance of 20% in the pre-instructional process. At the end of the instruction and probe sessions, it was seen that he responded correctly to the target stimulus with an average of 95%. In the follow-up sessions, which were conducted ten and twenty days later, it was seen that he maintained a level of 85%. The data related to the pass throwing skills of the subjects are given in Figure 2 below.

Yusuf showed an average performance of 15% in the pre-instructional process. At the end of the instruction and probe sessions, it was observed that he responded correctly to the target stimulus with an average of 95%. In the follow-up sessions, which were conducted ten and twenty days later, it was observed that he maintained 95%.

Ismail Can showed an average performance of 20% in the pre-instructional process. At the end of the instruction and probe sessions, it was observed that he responded correctly to the target stimulus with an average of 80%. In the follow-up sessions, which were conducted ten and twenty days later, it was observed that he maintained 80%.

Mikail showed an average performance of 20% in the pre-instructional process. At the end of the instruction and probe sessions, it was seen that he responded correctly to the target stimulus with an average of 85%. It was seen that he maintained a level of 90% in the follow-up sessions, which were conducted ten and twenty days later.

Salih Can showed an average performance of 20% in the pre-instructional process. At the end of the instruction and probe sessions, it was seen that he responded correctly to the target stimulus with an average of 90%. In the follow-up sessions, which were conducted ten and twenty days later, it was seen that he maintained a level of 90%. The data on the dribbling skills of the subjects are given in Figure 3 below.

Yusuf showed an average performance of 15% in the pre-instructional process. At the end of the instruction and probe sessions, it was observed that he responded correctly to the target stimulus with an average of 90%. In the follow-up sessions, which were conducted ten and twenty days later, it was observed that he maintained 90%.

Ismail Can showed an average performance of 20% in the pre-instructional process. At the end of the instruction and probe sessions, it was observed that he responded correctly to the target stimulus with an average of 90%. In the follow-up sessions, which were conducted ten and twenty days later, it was observed that he maintained 90%.

Mikail showed an average performance of 20% in the pre-instructional process. At the end of the instruction and probe sessions, it was seen that he responded correctly to the target stimulus with an average of 90%. It was seen that he maintained a level of 90% in the follow-up sessions, which were conducted ten and twenty days later.

Salih Can showed an average performance of 20% in the pre-instructional process. At the end of the instruction and probe sessions, it was seen that he responded correctly to the target stimulus with an average of 90%. In the follow-up sessions, which were conducted ten and twenty days later, it was seen that he maintained a level of 90%.

### Generalization Findings

Generalization sessions of the research were conducted using the preliminary test-end test model. Instructional sessions on generalization were held after all mass probe sessions. There was an interpersonal generalization study with sessions with the sports teacher who worked at the institution. In addition, these sessions were conducted in a different environment and generalization was made between environments. Yusuf, Ismail Can, Mikail, and Salih Can’s performances in generalization sessions are shown in Figure 4.

Looking at all the findings, Yusuf’s preliminary test session before teaching showed that his basic basketball skills were at a 15% level. The post-test session after teaching showed a 90% level of performance. The preliminary test session of Ismail Can before teaching showed that his basic basketball skills were at a 20% level. The post-test session after teaching showed a 90% level of performance. Mikail’s preliminary test session before teaching showed that his basic basketball skills were at a 20% level. The post-test session after teaching showed a 90% level of performance. Salih Can’s preliminary test session before teaching showed that his basic basketball skills were at 20% and his post-test session after teaching showed that he performed at a 90% level. When the opinions of the basketball teacher and the students were evaluated, it was determined that the students’ self-confidence and peer relations were strengthened thanks to their active participation in the lessons. The students were not bored because they stated that they found the lessons interesting and fun.

In light of these findings, teaching studies using the teaching of video modeling can conclude that subjects can generalize the skills they have gained in different environments and with different people.

## 4. Discussion

It was observed that the students who participated in the study showed great interest in computers and video screenings from the beginning of the instructional sessions in which computer-assisted video modeling instruction was used. Therefore, it is thought that students’ interest in technologies, such as computers and videos, and thus their high motivation has an effect on their acquisition of the targeted skills. This situation is similar to the findings obtained from other studies.

In the study conducted by Baglama, it was seen that teaching with a video model effectively taught subtraction and addition skills in mathematics [41]. In the study conducted by Gülsöz and Çıkılı, it was observed that video modeling was effective in teaching cold drink preparation and presentation skills [42]. As a result of the research conducted by Ertekin, Ece, Yıkmış (2015), it was seen that teaching with a video model was effective in teaching daily living skills [43]. As a result of the research conducted by Odluyurt (2013), it was seen that teaching with a video model was effective in teaching game skills [44]. As a result of the research conducted by Genç (2010), it was concluded that video modeling was effective in teaching food and beverage preparation skills [45], in teaching pasta cooking skills [46], and in teaching daily life and social skills and in gaining generalization skills [31,32,47,48,49,50,51,52].

## 5. Conclusions

In this research, the effect of basic basketball movements on the effectiveness and persistence of teaching of video modeling was investigated for four students with Down syndrome. According to the research findings, teaching by video model was determined to be effective in giving Down syndrome individuals basic basketball skills, generalized alongside different environments and different people, and persistence after 10–20 days.

Depending on the findings, the following recommendations have been made:With video, the effectiveness of teaching to model in teaching different types of obstacles can be explored.In skill teaching, video model and teaching method can be compared using a different teaching method.A different sports skill teaching effectiveness can be explored with the method of being a model by video.When curriculum development studies are carried out by curriculum development experts, studies suitable for teaching with a video model can be prepared.In future studies, researchers can look at the effectiveness of the viewpoint technique, which is a different technique from the video modeling teaching method, in teaching sports skills to disabled children.Teachers, experts who are working in the field, and families can be taught about how the video model should be implemented in teaching sports skills.

## Figures and Tables

**Figure 1 children-10-00153-f001:**
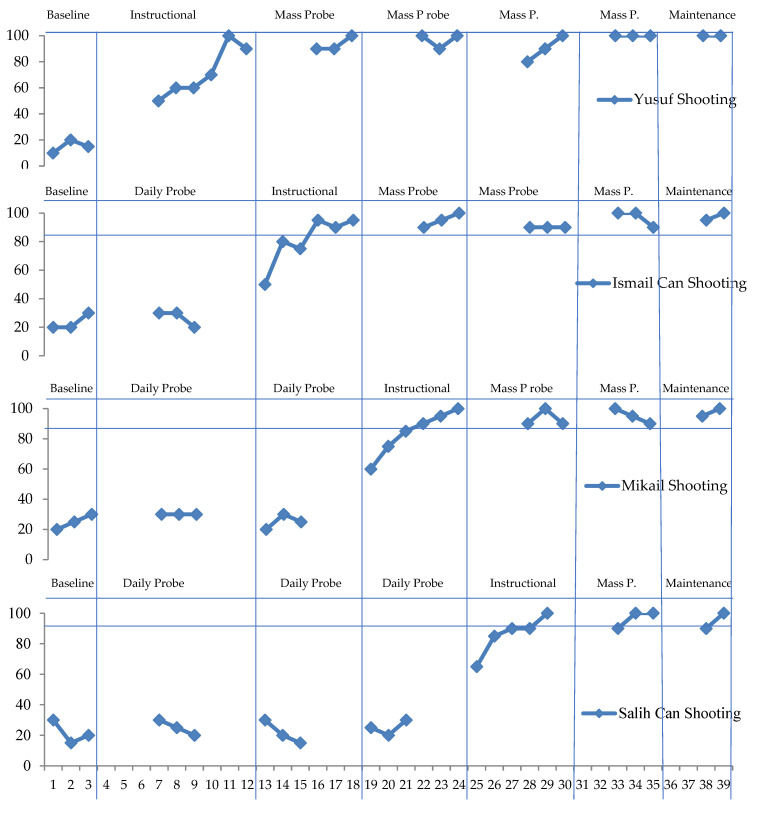
Data on the instructional of shooting skills in basketball maintenance.

**Figure 2 children-10-00153-f002:**
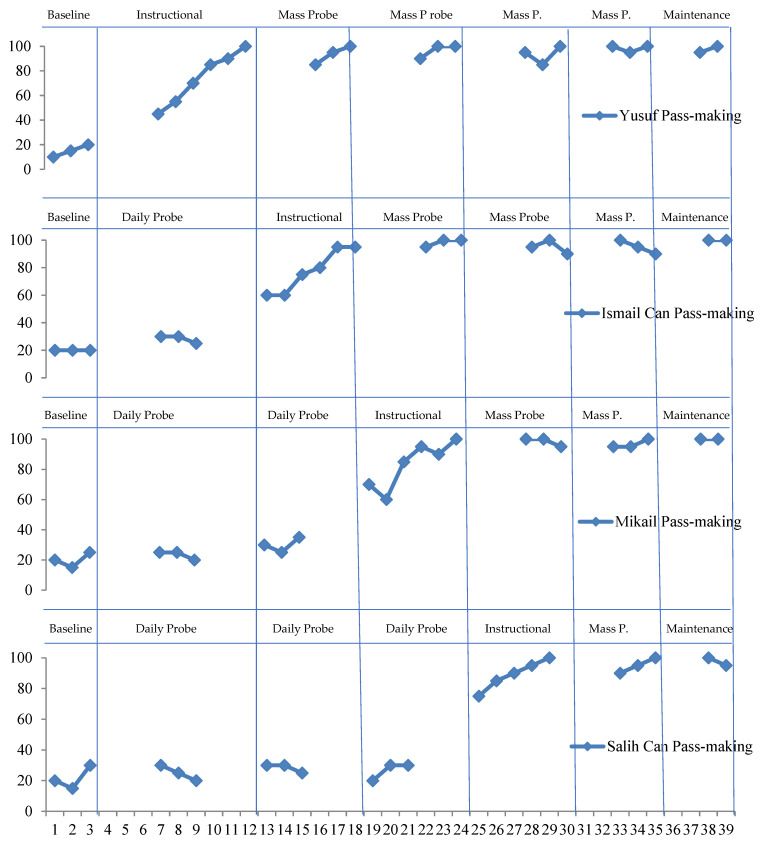
Data on instructional the ability to pass in basketball.

**Figure 3 children-10-00153-f003:**
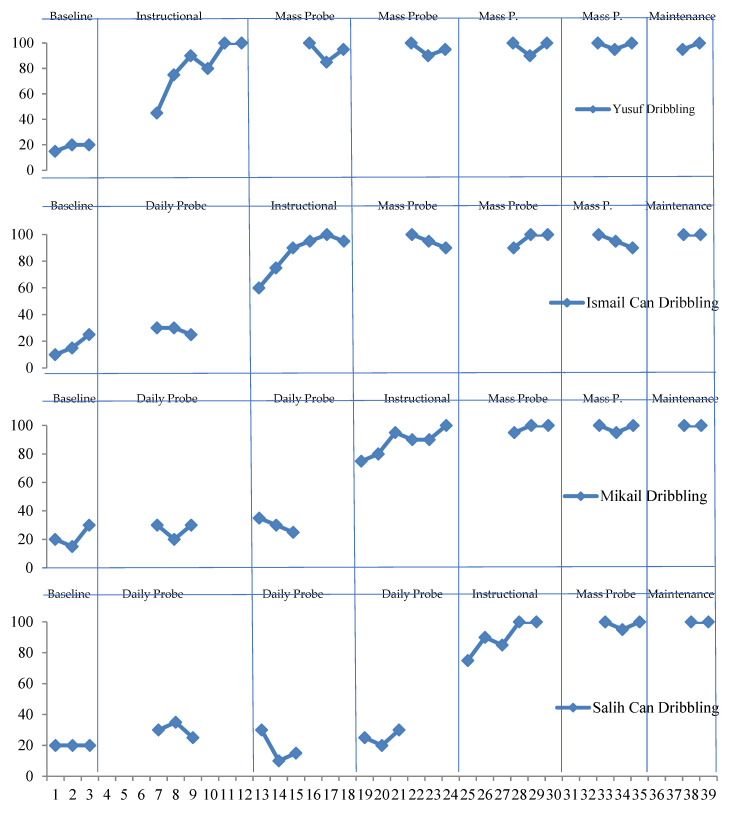
Data on the instructional of dribbling skills in basketball.

**Figure 4 children-10-00153-f004:**
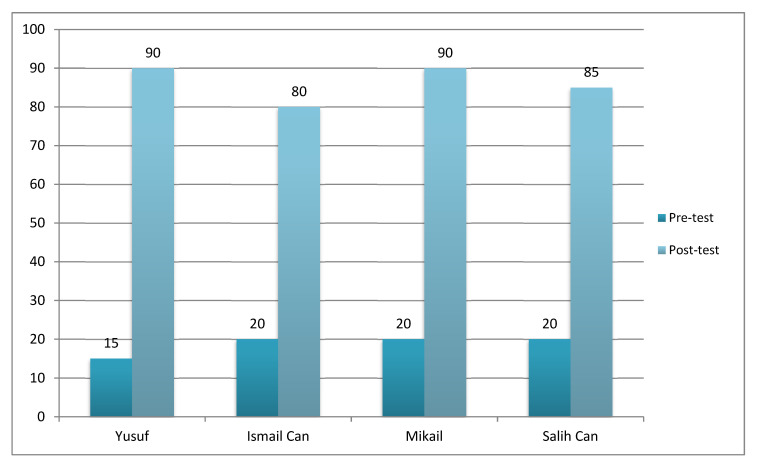
Basic basketball skills generalization data.

## Data Availability

Not applicable.

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
