# Peer review of "The Effectiveness of Computer Aided Video Modeling in Teaching Basic Basketball Movements to Individuals with Down Syndrome"

_children, 2023, doi:10.3390/children10010153_

Round 1
Reviewer 1 Report
Extensive editing of English language and style required.
It is hard to follow as it is now.
Many sentences are incomprehensible due to language errors or incorrect wording. I am under the impression that the research itself is sound. But extensive editing of language is required. For instance:
L26 “caused by the twenty-first (21) chromosome containing an extra chromosome” A chromosome does not contain a chromosome. BETTER: “caused by an extra copy of the twenty-first chromosome.”
L28 “Down syndrome is a surplus of a plus chromosome, seen mostly along with mental incompetence.” What is a surplus of a plus chromosome? And in English the term "mental incompetence" is not used in articles about people with intellectual disabilities. BETTER: “People with Down syndrome have some degree of intellectual disability ⎯ usually in the mild to moderate range.”
L30
“Special-needs individuals also require a variety of skills required by societal life, such as normal-development individuals.”
BETTER: “Just like individuals with a typical development, special-needs individuals need to learn a variety of skills required by societal life.”
In addition, the introduction should be further divided into paragraphs. Now, new topics are introduced but the paragraph continues without a break.
Author Response
Dear Reviewer,
First of all, thank you for your comments.
Necessary explanations have been made for your comments.
Details are in the attached file.
Best Regards,

Reviewer 2 Report
I advise the authors to remove point 2.2 (Practitioner);
its seems that The Effectiveness of Computer Aided Video Modelling is very usefull but the study is fragil because they are only 4 participants and its difficult to explore these results.
i suggese that the authors enlarge the sample and verify if the results trend is the same.
Author Response

(The authors gave the same response as above.)

Round 2
Reviewer 2 Report
i accept the authors corrections.